# The Patho-Neurophysiological Basis and Treatment of Focal Laryngeal Dystonia: A Narrative Review and Two Case Reports Applying TMS over the Laryngeal Motor Cortex

**DOI:** 10.3390/jcm11123453

**Published:** 2022-06-15

**Authors:** Maja Rogić Vidaković, Ivana Gunjača, Josipa Bukić, Vana Košta, Joško Šoda, Ivan Konstantinović, Braco Bošković, Irena Bilić, Nikolina Režić Mužinić

**Affiliations:** 1Laboratory for Human and Experimental Neurophysiology, Department of Neuroscience, University of Split School of Medicine, 21000 Split, Croatia; 2Department of Medical Biology, University of Split School of Medicine, 21000 Split, Croatia; igunjaca@mefst.hr; 3Department of Pharmacy, University of Split School of Medicine, 21000 Split, Croatia; jbukic@mefst.hr; 4Department of Neurology, University Hospital Split, 21000 Split, Croatia; vanakosta@gmail.com; 5Signal Processing, Analysis, and Advanced Diagnostics Research and Education Laboratory (SPAADREL), Faculty of Maritime Studies, University of Split, 21000 Split, Croatia; jsoda@pfst.hr; 6Department of Neurosurgery, University Hospital of Split, 21000 Split, Croatia; ivan.konstan@gmail.com; 7Department of Otorhinolaryngology, University Hospital of Split, 21000 Split, Croatia; bboskovic01@gmail.com (B.B.); ire.bilic@gmail.com (I.B.); 8Department of Medical Chemistry and Biochemistry, University of Split School of Medicine, 21000 Split, Croatia

**Keywords:** spasmodic dysphonia, laryngeal dystonia, dystonia, focal dystonia, focal laryngeal dystonia

## Abstract

Focal laryngeal dystonia (LD) is a rare, idiopathic disease affecting the laryngeal musculature with an unknown cause and clinically presented as adductor LD or rarely as abductor LD. The most effective treatment options include the injection of botulinum toxin (BoNT) into the affected laryngeal muscle. The aim of this narrative review is to summarize the patho-neuro-physiological and genetic background of LD, as well as the standard recommended therapy (BoNT) and pharmacological treatment options, and to discuss possible treatment perspectives using neuro-modulation techniques such as repetitive transcranial magnetic stimulation (rTMS) and vibrotactile stimulation. The review will present two LD cases, patients with adductor and abductor LD, standard diagnostic procedure, treatments and achievement, and the results of cortical excitability mapping the primary motor cortex for the representation of the laryngeal muscles in the assessment of corticospinal and corticobulbar excitability.

## 1. Introduction

The laryngeal motor cortex (LMC) plays a vital role in human voice and speech production. The functional organization of LMC and its interactions with other cortical (such as Broca’s area) and subcortical brain regions warrants further investigation due to still as of yet unsuccessful treatments of neurological voice disorders such as laryngeal dystonia (LD). To date, methodologies for mapping LMC with TMS [1,2,3,4,5,6] and intraoperatively by electrical stimulation (ES) techniques [5,6,7] have been previously developed to record corticobulbar motor evoked potentials (MEPs) from laryngeal muscles. Except for estimating the amplitude and latency of MEPs recorded from laryngeal muscles, the cortical silent period (cSP) was investigated from the thyroarytenoid muscle as a measure of LMC excitability in the TMS study [8]. Currently, it is thought that the cSP reflects an intracortical inhibitory process mediated by GABA_A_ and GABA_B_ receptors [9,10]. Previous work using TMS has indicated reduced inhibition to be characteristic of focal laryngeal dystonia (LD), hand dystonia, cervical dystonia, and spasmodic dysphonia (focal laryngeal dystonia) [11].

Focal LD is a rare, idiopathic disease affecting the intrinsic muscles of the larynx with a prevalence of 14–35 per 100,000 people, predominantly affecting women (4:1 ratio) [12,13], with average onset at around 40 years of age [14,15]. Vocal symptoms range from sporadic difficulty to sustained inability to phonate, with vocal tremors (voice breaks) or strained or choked speech. LD presents with two phenotypes, the more common adductor LD (adLD) and the relatively rare abductor LD (abLD). Although symptoms of these two types of LD differ, both are characterized by the loss of voluntary control of voice/speech production. Currently, there is no cure for LD, and the disease is often treated with botulinum toxin (BoNT), speech, and supportive voice therapy, and not frequently by using medication due to side effects [16,17,18,19]. Reliably good responses can be expected for the adductor LD with BoNT, with a reduction in voice breaks, reduction in speaking effort, and increased quality of life. However, BoNT therapy requires regular injections every three or several months to ensure the continuity of benefits. Patients often experience bothersome side effects, including pain from injections, breathiness, dysphagia, and hypophonia. A less common side effect of BoNT is dysphagia which can be severe [17]. 

This narrative review aims to summarize the knowledge on the patho-neurophysiological and genetic background of LD, standard recommended therapies, pharmacological treatment options, and the knowledge and promises of using neuro-modulation techniques such as repetitive transcranial magnetic stimulation (rTMS) and vibrotactile stimulation in the treatment of focal LD. The review will present two LD cases, patients with adLD and abLD, diagnostic procedure, and treatment achievement. 

## 2. LD Terminology, Speech Task Specificity, and Clinical Assessment

The group of multidisciplinary experts of the NIH/NIDCD Workshop on Research Priorities in Spasmodic Dysphonia/Laryngeal Dystonia (August 2019) adopted the term laryngeal dystonia (LD) instead of “spasmodic dysphonia”, and LD was recognized as a multifactorial, phenotypical heterogeneous form of isolated dystonia [13]. The isolated, focal LD is a rare neurological disorder of the laryngeal muscles affecting speech production while leaving whispering and innate and/or upper respiratory vocal behaviors such as crying, coughing, yawing, or laughing unaffected. The clinical assessment is challenging due to the lack of diagnostic biomarkers, and very often, the establishment of the diagnosis is delayed by approximately 4–5 years [13]. LD is frequently diagnosed by standard procedures, including: (a) endo-video-stroboscopic examination to evaluate vocal fold anatomy and movements during speech and other vocal activities of the larynx; (b) speech-language pathological examination assessing voice symptoms (including acoustic analysis); and (c) neurological evaluation for signs of regional dystonia, other movement disorders or any other neurological deficit (lesion). LD symptomatology is differentiated from dystonic vocal tremor, essential tremor, and muscle tension dysphonia based on LD task specificity [13]. Multidisciplinary experts from neurology, otolaryngology, speech-language pathology, neurosurgery, genetics, and neuroscience might be involved in establishing LD diagnosis or conducting research. 

## 3. LD Risk Factors

### 3.1. Genetic Risk Factors

The LD etiology remains unknown and is considered characteristically multifactorial. According to reported findings, up to 25.3% of LD patients have a family history of dystonia [13]. Hereditary dystonias are genetically and clinically heterogeneous. To date, genetic variants that have been studied among LD patients include mutations in the *TOR1A*, *TUBB4A*, *THAP1*, *ANO3*, *GNAL*, *SGCE*, *PRKRA*, *COL6A3*, and *KMT2B* genes [13,20,21] (Appendix A, [22,23,24,25,26,27,28,29,30,31,32,33,34,35,36,37,38,39,40,41,42,43,44]). The known genetic forms of LD include the mostly autosomal dominant mode of inheritance.

### 3.2. Extrinsic Risk Factors

Although there is no established link between focal LD to occur due to the causative influence of extrinsic factors, some health conditions and environmental agents might have a role as a trigger for LD [13]. Family history of dystonia and a history of psychological disturbances such as depression, anxiety, and stress might lead to a potential risk factor in developing LD [13]. Further, white females, as well as professionals using their voice more pronouncedly as teachers, speech and language pathologists, and singers, have been identified as having a higher risk of developing LD [13]. Underlying LD risk factors also include infections of the respiratory system, gastrointestinal diseases, and neck injuries [45].

## 4. Patho- Neurophysiology of LD

### 4.1. Neural Structures and Function

Although the pathophysiology of LD is not fully known, it has been suggested that LD is a functional and structural disorder involving a complex neuronal network comprising basal ganglia structures, the thalamus, and their connections with cortical areas, the cerebellum, and sensorimotor cortex [46,47,48,49,50,51,52]. Alterations in activity of speech-related areas mediating motor preparation and execution were reported in the primary motor cortex for oro-laryngeal muscle representation, the middle frontal gyrus, the inferior frontal gyrus (i.e., Broca’s area) [48,50,53,54,55,56,57], and the temporal [48] and parietal brain areas [58]. Further, the adductor and abductor laryngeal muscle movements are under the voluntary control of the corticobulbar tract projecting to the nucleus ambiguous of the brainstem. Alterations in the microstructural and functional integrity of the corticobulbar tract descending pathway from the primary motor cortex for representation of laryngeal musculature to the brain stem nuclei involved in voice/speech production might also be implicated in the pathophysiology of LD [51]. Processing of auditory and visual information during speech might also have a role in the pathophysiology of LD [55,56]. 

### 4.2. Knowledge of the Neurophysiological Basis of LD

Neurophysiological studies indicate altered inhibitory mechanisms in LD, as with cervical dystonia and focal dystonia of the hand. More precisely, cSP has been reported to be shortened in laryngeal thyroarytenoid muscle patients with adLD [8,11]. The cSP is measured as the duration of the electrical silence in the laryngeal muscle during vocalization and the simultaneous application of a single magnetic pulse with TMS at an intensity greater than the resting motor threshold (RMT) for the upper extremity hand muscle [8]. The cSP duration is a measure of GABA_B_-mediated inhibition of the motor cells of the primary motor cortex through the activity of inhibitory interneurons located within the superficial cortical layers of the primary motor cortex [8]. Decreased inhibition in the primary motor cortex may be due to dysfunction that may also occur in other cortical or subcortical areas that send projections to the primary motor GABA_B_ inhibitory interneurons [8]. Thus, decreased inhibition of GABA_B_ within the primary motor cortex may result from the dysregulation of neural circuits that consequently affect the balance of excitation and inhibition in the primary motor cortex [8]. There is no convincing and reliable evidence of changes in other neurophysiological measures such as the RMT or the active motor threshold in LD [8,11]. There are insufficient studies using TMS in patients with LD that could provide information on neurophysiological measures other than cSP, such as inhibitory measures of long-interval intracortical inhibition (LICI), short-interval intracortical inhibition (SICI), and short and long afferent latency inhibition (SAI, LAI). LICI is a paired-pulse technique with a conditioned magnetic pulse on intensity above the threshold and test stimuli (intensity below the threshold) applied in intervals between 50–200 ms leading to the suppression of cortical activity (suppression of motor evoked potential amplitude). SICI is a paired-pulse technique with a conditioned pulse below the threshold and test stimuli (above the threshold) applied in intervals between 1.5–2.1 ms leading to the suppression of cortical activity (suppression of motor evoked potential amplitude) [59,60]. There are no data for LICI and SICI in LD. SAI and LAI are techniques that can induce the suppression of motor evoked potential amplitude by applying an electrical pulse to the periphery (median nerve) followed by a magnetic pulse over the primary motor cortex (usually induced at an interstimulus interval of N20 ms + 2 for SAI and LAI is induced at an interstimulus interval of about 200 ms). SAI and LAI measures relate to GABA_A_ transmission [59,60,61]. So far, there is evidence of reduced afferent inhibition in focal dystonia of the arm or cervical dystonia [62,63,64,65].

Decreased afferent-induced inhibition indicates abnormal sensorimotor integration within the primary motor cortices, which is not surprising as it is known that the processing of sensory (sensorimotor) information in dystonia is altered. The presence of a sensory gesture (in cervical dystonia) also suggests abnormal reliance on sensorimotor networks and a potential mechanism for alleviating dystonic contraction. Understanding the mechanisms leading to reduced afferent-induced inhibition in isolated dystonia may provide new therapeutic goals that could be explored in future research to alleviate sensorimotor symptoms. Future neurophysiological studies with transcranial magnetic stimulation (TMS) should use more homogenous cohorts of adLd and abLD subjects and publish raw data values of corticobulbar motor evoked potentials from affected and non-affected laryngeal muscles, cSP, SICI, and SAI measures.

## 5. Focal LD Treatment Options

### 5.1. Standard Treatment with Botulinum Toxin (BoNT)

Botulinum toxin is a natural neurotoxin produced by the bacteria Clostridium botulinum that causes muscular paralysis. The primary mechanism of action of the toxin is via the inhibition of calcium-dependent exocytosis and the release of acetylcholine at the neuromuscular junction [66]. The effect of botulinum toxin is reversible because the nerve terminals recover the ability to release acetylcholine into the neuromuscular junction. Two types have been developed for clinical use in humans: type A has the longest duration of effect and diffuses less from the injection point compared with type B. The dosing differs significantly between type A and type B preparations. The most common type of botulinum toxin used in LD therapy is type A (Botox, Allergan, Irvine, CA, USA; Dysport, Ipsen, Ltd., Slough, UK). Adverse effects of botulinum toxin treatment may result from over-weakening of the intended target muscle and unintended weakening of the surrounding muscles. Therefore, both appropriate dosing and the tissue distribution of the toxin are crucial. In general, the dose is proportional to the targeted muscle mass, although the range of therapeutic dosing is typically highly variable [67]. Some patients get the best results from a unilateral dose and others from bilateral treatment. For example, in bilateral injections for adLD, therapeutic doses range from 0.3–15 U per thyroarytenoid muscle, although most adLD is well controlled with doses of 0.625–2.5 U [44]. The American Academy of Otolaryngology-Head and Neck Surgery (“AAO-HNS”) considers botulinum toxin a safe and effective modality for the treatment of LD, and it may be offered as primary therapy for this disorder. The goal of treatment is to give an injection that will provide just enough weakness to relieve spasm in the target muscles for as long as possible without causing unnecessary weakness in neighboring muscles resulting in dysphagia, and prolonged breathiness (adductor), or airway compromise (abductor) [68]. There are a variety of injection approaches to deliver botulinum toxin to the larynx: percutaneous injection with EMG guidance (most traditional), percutaneous with laryngoscopic guidance, and supraglottic botulinum toxin injection with laryngoscopic guidance. For adLD, the intrinsic laryngeal injection muscles are the thyroarytenoid, lateral cricoarytenoid, and interarytenoid muscles. These muscles can all be accessed through the cricothyroid membrane. For the thyroarytenoid muscle, it is helpful to bend the needle upward to 30–45°. The needle is inserted through the skin either at or just off the midline. The needle tip is then directed superiorly and laterally, advancing towards the ipsilateral thyroarytenoid muscle. The cricothyroid membrane is palpated to inject into the lateral cricoarytenoid muscle, and the needle is placed through the cricothyroid membrane in this location and is angled superiorly. Further, the lateral cricoarytenoid muscle is more lateral than the thyroarytenoid muscle and is encountered more superficially. For abLD, the posterior cricoarytenoid muscle can be accessed anteriorly by piercing through the cricoid rostrum or laterally by rotating the larynx. The lateral approach to the cricoarytenoid muscle requires a relaxed patient, preferably with a relatively thin neck. The patient must tolerate the clinician applying moderate pressure/force on their larynx to rotate the posterior aspect of the cricoid into a position to allow access. The needle is inserted traversing the pyriform sinus and inferior constrictor, then it is further advanced until it stops abruptly against the cricoid cartilage’s rostrum [69,70].

### 5.2. The Long Term Effects of BoNT

Although botulinum toxin is generally considered safe, its widespread use and the constantly expanded indications raise safety issues. In February 2008 and April 2009, the Food and Drug Administration (FDA) published an early communication regarding botulinum toxin type A and botulinum toxin type B, informing physicians that these drugs have been associated with systemic adverse reactions, including respiratory compromise and death resembling those seen with botulism, in which botulinum toxin spreads to the body beyond the injection site [71]. In 2005, the FDA raised safety issues regarding botulinum toxin in a published analysis of adverse events covering the period from 1989 to 2003. According to that publication, there were 407 adverse event reports related to the therapeutic use of botulinum toxin (median dose of 100 units), 217 of which met the FDA’s definition of serious adverse events. Few data on the long-term adverse events of botulinum toxin were identified. Most of them concern the therapeutic use of botulinum toxin. Long-term safety data indicate that toxic effects of botulinum toxin can appear at the 10th or 11th injection after prior uncomplicated injections. The longest follow-up study of 45 patients continuously treated with botulinum toxin for 12 years identified 20 adverse events in 16 patients, including dysphagia, ptosis, neck weakness, nausea/vomiting, blurred vision, marked weakness, chewing difficulties, hoarseness, edema, dysarthria, palpitations, and general weakness [72]. Diffusion of botulinum toxin to contralateral muscles has also been reported. Animal studies have shown that botulinum toxin can spread to a distance of 30–45 mm from the injection site [72]. However, generalized diffusion of botulinum toxin is possible, especially after long-term therapeutic or cosmetic use. The effects of generalized diffusion are not well studied. The mechanism responsible for the generalized diffusion of botulinum toxin is not known. Proposed hypotheses concern either a systemic spread or a retrograde axonal spread of the toxin. Systemic toxin spread can lead to adverse events suggesting botulism, including muscle weakness or paralysis, dysarthria, dysphonia, dysphagia, and respiratory arrest.

Additionally, experimental studies in rodents have shown that botulinum toxin receptors exist in the central nervous system, and a small amount of botulinum toxin crosses the blood-brain barrier [73]. This raises the possibility that botulinum toxin is transported in a retrograde manner, similar to tetanus toxin, and may cause centrally mediated side effects [74]. Davidson and Ludlow [75] studied whether physiological changes can be found in laryngeal muscles following repeated treatment with botulinum toxin injections in spasmodic dysphonia. Seven patients whose treatment consisted of multiple unilateral thyroarytenoid injections were examined more than six months following their most recent botulinum toxin injection by fiberoptic laryngoscopy and electromyography. Comparisons were made between injected and contralateral noninjected muscles’ motor unit characteristics, muscle activation patterns, and vocal fold movement characteristics. The results demonstrated that motor unit characteristics differed between injected and noninjected muscles and that these differences were more significant in patients less than 12 months since the last injection. Motor unit duration differences were reduced, and motor unit amplitude and numbers of turns were increased in muscles sampled over one year after injection. These results suggest that while the physiologic effects of botulinum toxin are reversible, the re-innervation process continues past 12 months following injection [75]. Repeated injections may eventually enhance the pathological innervation, leading to tolerance and even exacerbation of local symptoms.

Moreover, they cause muscle fibrosis after several years, though such an effect has not been shown in shorter follow-ups so far [76] Resistance to botulinum toxin due to the development of antibodies to the toxin has also been reported as a long-term adverse event of the therapeutic use of botulinum toxin [76]. Immunoresistance develops within the first years of therapy. It is unlikely to develop if immunoresistance to botulinum toxin is not noted within the first four years.

### 5.3. Review of the Literature on BoNT and Deep Brain Stimulation (DBS) Treatment

Table 1 presents findings of BoNT treatments of LD and an overview of the literature on invasive brain modulation with deep brain stimulation (DBS) [77,78,79,80,81]. The efficacy and safety of BoNT were established for the treatment of LD, and this approach is considered by most to be the treatment of choice for spasmodic dysphonia/LD, particularly adLD [16]. Most studies report about 75–95% improvement in voice symptoms after BoNT [81,82]. Invasive brain stimulation with the DBS of unilateral or bilateral globus pallidus internus (GPi) or subthalamic nucleus (STN) has been approved by the FDA for the treatment of drug-refractory generalized, segmental, and cervical dystonias and hemidystonia [13,83,84], as well as for the treatment of essential tremor in adult patients whose tremor is not adequately controlled with medication. Table 1 presents patients with essential tremor and LD treated with DBS [78,79,80]. Generally, it is agreed that LD may show a poorer response to DBS [85,86].

### 5.4. Pharmacological Treatment Possibilities and Effectiveness in Dystonia Treatment

The most commonly used dystonia treatment, BoNT, has some limitations, e.g., it is painful for patients and can cause swallowing difficulties. Therefore, there is still an unmet need for effective dystonia pharmacological treatment [62]. The currently available pharmacological treatment involves medicines that act on gamma-aminobutyric acid (GABA), dopamine, or acetylcholine neurotransmitter pathways. Furthermore, novel treatments are also mainly focused on the same neurotransmitter pathways central in dystonia pathophysiology. However, most of the widely used drugs among dystonia patients still have low levels of efficacy evidence [87,88]. 

Trihexyphenidyl, one of the most commonly used anticholinergic drugs, is the treatment of choice for childhood-onset dystonia, as it has been usually well tolerated in this patient group. It could also be used in adults. The daily dose should be determined empirically but it most commonly ranges between 5–15 mg, though, if tolerated, dosages could be much higher (100 mg is the maximal daily dose recommended). The initial dosage is usually 1 mg, and then it should be increased by 2 mg every 3–7 days divided into three daily doses. The major concern regarding its use is the possibility of trihexyphenidyl to increase intraocular pressure which leads to vision blurring and possibly narrow-angle glaucoma. Other not so uncommon adverse reactions are sedation, memory impairment, psychosis, chorea, blurred vision, urinary retention, constipation, and dry mouth. Levodopa, in combination with carbidopa, an inhibitor of aromatic amino acid decarboxylation, is a widely used dopaminergic drug in dystonia patients. The dose and titration are similar to their use in mild Parkinson’s disease (slow titration till daily doses of 300–400 mg of levodopa divided in three doses, starting with 50 mg of levodopa). The most common side effects are low blood pressure, nausea, confusion, and dyskinesia. Lastly, as adjunctive therapy, GABA agonists are used to relaxing muscles in dystonia patients. The most commonly used benzodiazepines in LD patients are clonazepam, diazepam and lorazepam [87]. A maximal recommended daily dose of clonazepam is 4 mg divided into 2–3 doses. The start is usually with 0.25–0.5 mg 2–3 times a day, and then the dosage is slowly increased every 3–5 days to 0.5 mg. The most common side effects are sedation, depression, nocturnal drooling, and behavioral disinhibition. Caution must be taken because abrupt discontinuation can trigger seizures. There are several other potential dystonia treatment alternatives described in the literature. The first one would be a medication that acts as vesicular monoamine transporter 2 inhibitors (VMAT2). The well-known representative of this medication group is tetrabenazine. Furthermore, the other medication groups are as follows: sodium oxybate; antihypertensive medication clonidine; antiepileptic’s gabapentin; zonisamide; antidepressant escitalopram, a selective serotonin reuptake inhibitor; and hypnotic medication zolpidem. Future therapies of dystonia should involve gene therapy aimed at the specific genes of dystonia patients [88,89]. 

Sodium oxybate is a sodium salt of g-hydroxybutyric acid used to treat narcolepsy, excessive sleepiness, and disturbed nighttime sleep. A study by Simonyan et al. [90] suggests that this medication has direct modulatory effects on abnormal neural activity of the dystonic network. This medication can raise blood pressure values due to the sodium content. However, research has shown a low frequency of cardiovascular adverse drug reactions and no association with cardiovascular risk [91]. Other than the cardiovascular risk in general, due to the symptoms of the disease, dystonia patients are prone to anxiety and depressive comorbidities, and the use of escitalopram could seem reasonable in patients with existing symptoms [92].

### 5.5. Future Neuromodulation Treatment Options and Vibrotactile Stimulation 

The effectiveness of repetitive transcranial magnetic stimulation (rTMS), a non-invasive neuromodulation technique, in assessing cortical excitability and inhibition of laryngeal musculature might be one of the potential treatment options for LD. Previous neurophysiological findings demonstrated decreased intracortical inhibition in patients with adLD compared to healthy controls [8,11]. Application of low frequency (inhibitory) rTMS to the LMC might decrease the over-activation of the laryngeal muscles [93]. Given that adductor LD has been found to be associated with decreased cortical inhibition and that 1 Hz is known to increase intracortical inhibition, the purpose of the pilot study by Prudente et al. [93] was to examine the effects of 1200 pulses of 1 Hz rTMS delivered to LMC in people with adductor LD and healthy individuals. This is the first feasibility study testing effects of 1 Hz rTMS in LD. The authors tested only a single session of 1 Hz rTMS and observed acoustical measures changes pointing to beneficial effects on voice symptoms. Future studies would need to test the long-treatment duration of 1 Hz rTMS in LD. To test this hypothesis of the beneficial effects of five days of treatment of 1 Hz rTMS, a proof-of-concept, randomized study was recently registered on 27 October 2021 (ClinicalTrials.gov Identifier: NCT05095740, accessed day: 20 May 2022), and the estimated study completion date is 31 May 2025/2026. 

Another non-invasive brain stimulation technique, transcranial direct current stimulation (tDCS), has not been reported in the assessment of LD according to the current state-of-the-art.

Further, a feasibility study was performed using vibrotactile stimulation (VTS) to treat LD [57]. The authors reported that 29 min of VTS in a one-day session improved the voice quality parameter (*smoot cepstral peak prominence*). Although a stimulation protocol by using VTS has been published, the optimal stimulation protocol for the treatment of LD is not yet known. After publishing a paper on a feasibility study of VTS, the authors started a clinical study, which is still ongoing, testing the effect of VTS for four weeks in patients with LD (ClinicalTrials.gov Identifier: NCT03746509, accessed day: 20 March 2022). The results have not yet been published, nor is their VTS stimulator commercially available. The same research group applied for a patent (the United States Patent Application Publication, Konczak et al., Pub. No. US 2019/0159953 A1, Pub. Date: 30 May 2019) where they protected the VTS solution of placing the vibrators on the laryngeal muscles over the skin in the form of a necklace placed around the neck.

## 6. Case Reports of LD Patients

### 6.1. Patient with adLD

#### 6.1.1. Clinical Findings 

A 55-year-old right-handed woman, a psychologist, started to present hoarseness in March 2015 (Appendix A). The first endoscopic examination (Karl Storz) (April 2015) confirmed laryngitis, spindle-shaped thickening of the vocal cords, decreased stroboscopic amplitudes, and prolonged adduction of the vocal cords. Voice saving therapy, speech therapy, a light diet, and taking Iberogast^®^ (Bayer AG, Kaiser-Wilhelm-Allee 1, 51373 Leverkusen, Germany) to improve digestion were recommended. The second examination (April 2016) confirmed dysphonia with thinner vocal cords, reduced Bernoulli effect, spasms during vocalization, extended closing phase, and reduced glottal wave. The vocal spasm was also detected during counting, muttering, and minor spasms in buzzing. Brain magnetic resonance imaging (MRI) was performed in October 2016 with normal finding. At the subsequent examination (January 2017), the dysphonia spastica was diagnosed. Spectral and multidimensional acoustic voice analysis showed that the spasm was partially reduced with prolonged phonation of vowel /i/ with high-frequency Fo (346 Hz). The vocal spasm was present in all verbal and vocal tasks except in whisper counting. Acoustic parameters of diadochokinesis (pa-pa) indicated a markedly long syllable duration and accelerated pronunciation change, as well as increased syllable variation. There were marked variations in frequency and amplitude in the analysis of vowel /a/ related to the quality of the voice and the frequency of tremors. Focal LD was diagnosed in February 2017. The first BoNT treatment with Dysport^®^, Galderma Laboratories, L.P., El Segundo, CA, USA (abobotulinumtoxinA) (15 units) was injected into the right vocal cord under electromyography (EMG) guidance. The second BoNT treatment was performed with Dysport^®^ (abobotulinumtoxinA) (15 units) injected in both vocal cords under EMG. Therefore, the patient was treated with BoNT in 2017 with a short-term improvement of up to ten days with swallowing difficulties and refused further treatment. The brain MRI was again performed in March 2021 with a normal finding. 

#### 6.1.2. Evaluation of Corticobulbar and Corticospinal Excitability with Transcranial Magnetic Stimulation (TMS) 

The corticospinal excitability measures (RMT, amplitude, and latency of motor evoked potentials for upper extremity muscles) and corticobulbar excitability measure (motor evoked potential latency from cricothyroid muscle) performed with single pulse transcranial magnetic stimulation (TMS) over the primary motor cortex (Nexstim NBS System 4 of the manufacturer Nexstim Plc., Helsinki, Finland) [4,94,95]. The MRI of the subject’s head was performed with Siemens Magnetom Area having Tim (76 × 18) of strength 1.5 T. MRI images were used for the 3D reconstruction of individual brain anatomy. With the subject comfortably seated, the MRI is co-registered to the subject’s head using the tracking system with Nexstim’s unique forehead tracker. The eight-shaped magnetic coil was used, generating a biphasic pulse with a length of 289 µs. The coil with an inner winding diameter of 50 mm and an outer winding diameter of 70 mm was placed tangentially to the subject’s skull over the primary motor cortex. The maximum electric field strength measured 25 mm below the coil in a spherical conductor model representing the human head was 172 V/m. The cSP was tested as an inhibitory cortical measure [94]. 

For recording the responses from the cricothyroid muscle, two hook wire electrodes (type 003-400160-6) (SGM d.o.o., Split, Croatia) were inserted into the cricothyroid muscle according to published methodology [4,96]. Surface electromyography electrodes (Ambu Blue Sensor BR, BR-50-K/12) were attached in a belly tendon fashion over the right APB muscle with the ground electrode over the dorsal surface of the APB muscle. Before insertion of the electrodes individually, the subject needs to slightly extend the neck and produce a high-pitch sound (i.e., /iiii…/). During this slight facilitation, it is helpful to palpate the contracted cricothyroid muscle belly between the thyroid and cricoid cartilages by marking this spot with the marker. Each hook wire electrode consists of Teflon-coated stainless steel wire 76 µm in diameter, passing through 27-gauge needles (0.4 mm), 13 mm in length. The recording wires have a stripped Teflon isolation of 2 mm at their tip and are curved to form the hook for anchoring them. The sampling rate was 3 kHz per channel, resolution 0.3 µV, scale −7.5–7.5 mV, CMRR > 90 dB, noise < 5 µV peak-to-peak, and frequency band 10–500 Hz. The RMT intensity for the upper extremity muscles (abductor pollicis brevis) was 49% of maximal stimulator output. Figure 1 (A)(B) presents positive cortical spots for primary motor cortical representation for upper extremity APB muscle and cricothyroid muscle with the recording of MEPs from APB and cricothyroid muscle. The single magnetic pulse intensity over the LMC was gradually increased from 51% to 80 % of maximal stimulator output that the subject could tolerate (reporting pain and discomfort due to activation of temporal musculature). Figure 2 shows application of a single magnetic pulse during vocalization, inducing motor evoked potential from the left cricothyroid muscle (latency of 11.3 ms) with no cSP induced in the cricothyroid muscle at 80 % of maximal stimulator output. 

#### 6.1.3. Electroneuronographic (ENG) Assessment of Motor and Sensory Nerves of Upper and Lower Extremity Muscles 

There were no deviations in electroneurographic (ENG) measures for upper and lower extremity muscles. ENG assessment of lower and upper extremities included the following measures for motor nerves (n. peroneus and n. tibialis: distal motor latency, compound muscle action potential amplitude, compound muscle action potential duration, conduction velocity, and F-wave latency); and for sensory nerves (n. medianus and n. ulnaris: sensory nerve action potential amplitude, sensory nerve action potential latency, and conduction velocity). The electrophysiological examination was performed using the Medelec-Synergy EMG instrument (Oxford Instrument Co., Surrey, UK). 

#### 6.1.4. Blood—DNA Analysis 

Routine blood analysis, including white blood count, erythrocyte sedimentation rate, and C-reactive protein, were within the normal range. There were no abnormal findings in iron, manganese, parathyroid hormone, and serum homocysteine level. A blood sample was collected, and the DNA was extracted from dried blood spots on filter cards (CentoCard^®^) using standard, spin column-based methods, following the manufacturer’s instructions. Targeted sequencing of the patient’s DNA was performed using a next-generation sequencing (NGS) panel, including 88 dystonia-associated genes (Centogene, Rostock, Germany). Genomic DNA was enzymatically fragmented, and Illumina adapters were ligated to generate fragments for subsequent sequencing on the NovaSeq 6000 platform (Illumina), with the average coverage targeted to at least 100× or at least 99.5% of the target DNA covered 20×. All coding regions of the panel genes, 10 bp of flanking intronic sequences, and known pathogenic/likely pathogenic variants within these genes (coding and non-coding) were targeted for the analysis. Data analysis, including alignment to the hg19 human reference genome (Genome Reference Consortium GRCh37), variant calling, and annotation, were performed using validated in-house software. No clinically relevant variants, including copy number variations (CNVs), were identified in the panel genes.

#### 6.1.5. Pharmacological Treatment Attempts 

Medication treatment with trihexyphenidyl (Artane, anticholinergic drug) was introduced on 20 March 2021, with a dosage of 2 mg daily per six days and increasing by 2 mg every six days. The patient reached 8 mg and ended the treatment after two weeks due to severe side effects (red eyes, anxiety, distractibility, lethargy) and with no signs of voice symptoms improvement. Medication treatment with benzodiazepine (clonazepam) (Rivotril, Roche) was introduced on 24 June 2021, with a dosage of 0.5 mg daily per six days and increasing by 0.5 mg every six days. In four weeks of treatment, the patient reached 2 mg with side effects (drowsiness, fatigue, sadness, crying). The patient monocyte subsets (anti-inflammatory, inflammatory markers) were no different before and after treatment with a benzodiazepine. In the acoustic analysis of vowel /a/ there were still distinctive variations in the frequency of tremors after benzodiazepine medication.

#### 6.1.6. Patient with adLD Conclusion 

The patient had severe dysphagia, which necessitated discontinuation of BoNT treatment and beginning treatment with drugs recommended to treat dystonia [16,18,97]. The medication treatment with an anticholinergic drug (trihexyphenidyl) was classified as the first-line agent for symptomatic therapy in dystonia, however, no findings were reported for treatment of the LD. The adLD patient in our study could not reach the recommended daily therapeutic level of 15 mg of trihexyphenidyl due to severe side effects. Benzodiazepine (clonazepam) was introduced as the second-line agent, but still, no effect was noticed on acoustic voice measures (Figure 3A,B). The patient case also showed that there is no genetic basis for LD disease, and cSP could not be recorded in laryngeal muscle due to lack of inhibition or insufficient intensity (80% of maximal stimulator output was rather high over the lateral part of LMC providing discomfort to the subject). Previous studies reported inducing cSP ranging from 48% to 72% of the maximal stimulator output in adLD subjects and in healthy subjects from 50% to 67% of maximal stimulator output [11]. The case report also shows the problem in the length of time to make a final diagnosis. In this particular case, it took two years to make a valid diagnosis. The patient is under consideration for an experimental trial with rTMS and tDCS.

### 6.2. Patient with abLD

#### 6.2.1. Clinical Findings

A 57-year-old male, by profession lawyer (judge) and singer of traditional Croatian a cappella singing, has had LD of the left vocal cord for three years. The patient noticed changes in his voice in January 2019 while singing in lower tones, and in June 2019, a breathless voice developed. LD was confirmed by endo-video-stroboscopy, acoustic voice analysis, and neurological evaluation for signs of regional dystonia, other movement disorders, or any other neurological deficit (lesion). Magnetic resonance imaging (MRI), and electroneuronography of upper and lower extremities revealed a normal finding, and electromyography confirmed possible spasms in the left cricothyroideus muscle. Although the treatment of choice for the patient’s conditions was botulinum toxin treatment, the patient underwent a vocal cord augmentation procedure as a second opinion from a private practice otolaryngologist. Autologous fat vocal fold augmentation is a general surgical procedure used to repair glottal incompetence in patients with unilateral vocal fold paralysis. Autologous fat is harvested from the lower abdomen, and the small fat grafts are purified from other tissues. Under microscope control, the autologous fat is injected into the thyroarytenoid muscle using an applicator with a special gear mechanism. Spectral and multidimensional acoustic voice analysis showed voice breaks predominant in speech tasks (i.e., vocalization of sound /i/—number of voice breaks 5, Jitter (local) (%)—5.45, Shimmer (local, dB)—1.64), and partially reduced when coughing (number of voice breaks 2, Jitter (local) (%)—2.14, Shimmer (local, dB)—1.34) and singing (i.e., vocalization of high pitch sound /i/—number of voice breaks 3, Jitter (local) (%)—1.90, Shimmer (local, dB)—1.39).

Pneumo-phonic voice, calcification in the left cricoarytenoid joint, and lagging of the left vocal cord in adductor movements were indicators for conducting the phono-surgical intervention (November 2019). However, the autologous fat injection for medialization of the left vocal fold did not improve the voice symptoms. The recommended medication therapy included propranolol, coenzyme Q10, vitamin B1 disulfide, vitamin B6, vitamin B12, and magnesium. The patient refused medication treatment recommended for dystonia treatment [16,18,97].

#### 6.2.2. Evaluation of Corticobulbar and Corticospinal Excitability with Transcranial Magnetic Stimulation (TMS)

The TMS technique was used for mapping the primary motor cortex for upper extremity hand and laryngeal muscle representation with a recording of RMT for upper extremity hand muscle (APB), motor evoked potentials from hand muscle (APB) at RMT, corticobulbar motor evoked potentials from laryngeal (cricothyroid muscle), and cSP from the cricothyroid muscle [4,8,11]. To facilitate the corticobulbar motor evoked potentials from the cricothyroid muscle to induce cSP, the subject vocalizes high pitch sound /i/ while slightly increasing the stimulation intensity starting from the referent RMT. RMT for the left hemisphere was 35% of maximal stimulator output, while the RMT intensity was 36% for the right hemisphere. Prolongations in cSP were detected in the right cricothyroid muscle (89.54 ± 21.9 ms) compared to cSP in the left cricothyroid muscle (44.78 ± 6.9 ms) [8] (Figure 4). The intensity of the left hemisphere primary motor cortex for cricothyroid muscle representation for cSP eliciting was of 63% of maximal stimulator output, while for the right hemisphere, it was 65% of maximal stimulator output.

#### 6.2.3. Patient with adLD Conclusion

This case illustrates a rare case of abLD who did not benefit from the autologous fat injection. We have provided the first results of TMS application in evaluating the neurophysiological measure of cortical inhibition such as cSP in abLD. The patient is under consideration for BoNT treatment and an experimental trial with rTMS and tDCS.

## 7. Discussion

Diagnosis and treatment of LD remain challenging as underlying patho- and neurophysiology are unclear and require further studies. Currently, there is no cure for LD, and the disease is often treated with BoNT, speech and voice supportive therapy, and rarely by using medication due to side effects [16,17,19,97]. BoNTs are widely used for the treatment of LD [16,17,19]. Reliably good responses can be expected for the adLD with BoNT, reducing voice breaks and speaking effort and increasing quality of life [19]. However, BoNT therapy requires regular injections every three or several months to ensure continuity of benefits. Additionally, patients often experience bothersome side effects, including pain from injections, breathiness, dysphagia, and hypophonia. A less common side effect of BoNT is dysphagia which can be severe [17], as we reported in our case report of a patient with adLD. Through extensive research of the literature, it was found that LD is mostly treated with BoNT (56.6%), but rarely with other medications [97]. Pirio Richardson et al. [97] reported that benzodiazepines were used to treat 16 patients with LD, while muscle relaxants were used in 3 patients, dopaminergic drugs in 3 patients, and non–benzohypnotic drugs in 4 patients. According to a study by Pirio Richardson et al. [97] baclofen and anticholinergic drugs were not reported to be used in the treatment of LD. Even though the medication treatment with an anticholinergic drug (trihexyphenidyl) was classified as the first-line agent for symptomatic therapy in dystonia [18], no findings were reported so far regarding the medication attempts for LD [18]. The adLD patient in our study could not reach the recommended daily therapeutic level of 15 mg of trihexyphenidyl due to severe side effects. Benzodiazepine (clonazepam) was introduced as the second-line agent [18], but no effect was noticed on acoustic voice measures. Both adLD and abLD patients in this report are currently with unchanged voice status, and further experimental trials are considered, such as rTMS or tDCS using inhibitory protocols such as 1 Hz [70].

Finally, the assessment of cSP with TMS in our two case reports of adLD and abLD patients points to altered intracortical inhibition mechanisms in LD, which is similar to the findings of Chen et al. [8,11]. The presented case of a male subject with abLD is an extremely rare type of LD, and according to our knowledge, it is the first case with a neurophysiological evaluation of cSP. Future studies with TMS are critical, involving a higher number of LD patients (both adLD and abLD types) investigating cSP from laryngeal muscle, as well as other neurophysiological measures such as SICI, LICI, and SAI [36,37,38,94]. A wider understanding of the neurophysiological basis of LD might lead to more efficient treatments, potentially involving noninvasive neuromodulation techniques such as rTMS or tDCS or vibrotactile stimulation of the laryngeal muscles.

The limitation of the present study relates to the neurophysiological assessment of the corticobulbar excitability by recording MEPs and assessment of cSP from non-targeted laryngeal muscles affected by LD disease. Future studies can adopt the procedures for the percutaneous introduction of recording electrodes into the target laryngeal muscles (i.e., thyroarytenoid muscle) affected by the LD [8,11].

## 8. Conclusions

Although LD diagnosis has improved, it remains unacceptably delayed [98,99], which is evident in our presented case of a female with adLD. Developed methodologies for mapping the corticobulbar pathway [4,5,6,7,8,11] provide a tool for assessing neurophysiological measures such as MEP responses from laryngeal muscles and cSP in LD patients. The recent neurophysiological studies with TMS point to the impaired intracortical inhibition measured with a non-invasive cSP measure. The current lack of full understanding of LD etiology and patho–neurophysiology contributes to limited therapeutic interventions, but hopefully, promising neuromodulatory techniques such as rTMS might bring new light to the treatment of LD disorder.

## Figures and Tables

**Figure 1 jcm-11-03453-f001:**
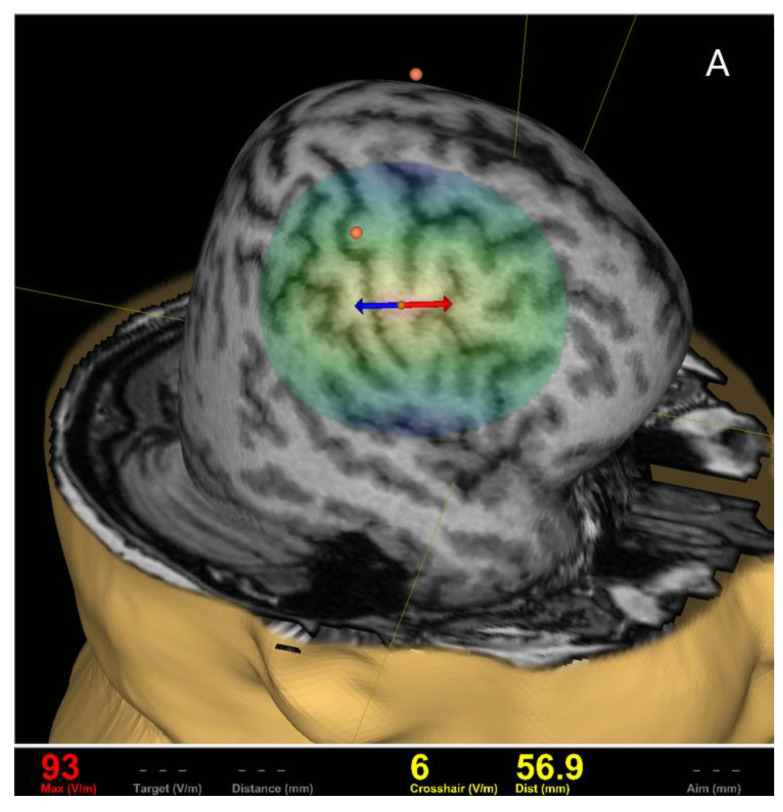
Primary motor cortical representation for the upper extremity left APB muscle and left cricothyroid muscle (**A**) with the recording of MEPs from APB and the cricothyroid muscle (**B**). Note: The orange spot on(**A**) depicts the cortical spot for APB muscle and MEP recording from APB muscle (**B**, upper channel, green color), and the orange spot with the position of the magnetic coil over the primary motor cortex (LMC) denotes the positive spot for inducing MEP in the cricothyroid muscle (lower channel on **B**, pink color). The latency of MEP in the cricothyroid muscle is 13.3 ms, and the amplitude of 863 µV, while the MEP latency of APB is 23.7 ms, and an amplitude of 228 µV. The stimulation intensity was 80% of maximal stimulator output with the subject engaged in phonation task (the left traces on **B** depict laryngeal muscle contractions in free-running electromyography).

**Figure 2 jcm-11-03453-f002:**
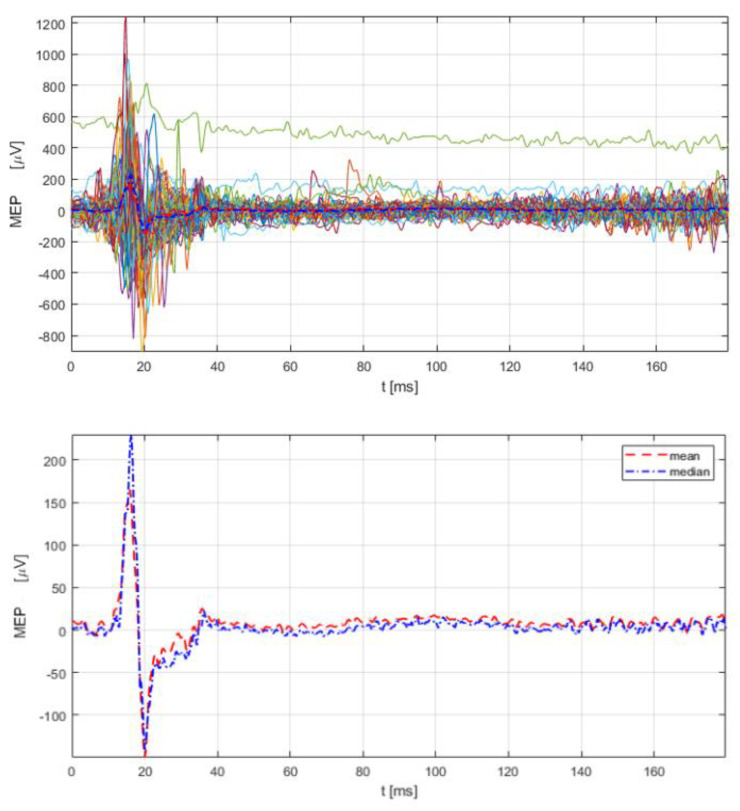
Mapping of the LMC (right hemisphere) and recording corticobulbar motor evoked (MEPs) from the cricothyroid muscle at intensities of 60–80% of the maximal stimulator output and with MEP latency of 11.3 ms. On the upper part of the figure are overlapped MEPs, and on the lower part are the mean and median of these responses. cSP could not be recorded.

**Figure 3 jcm-11-03453-f003:**
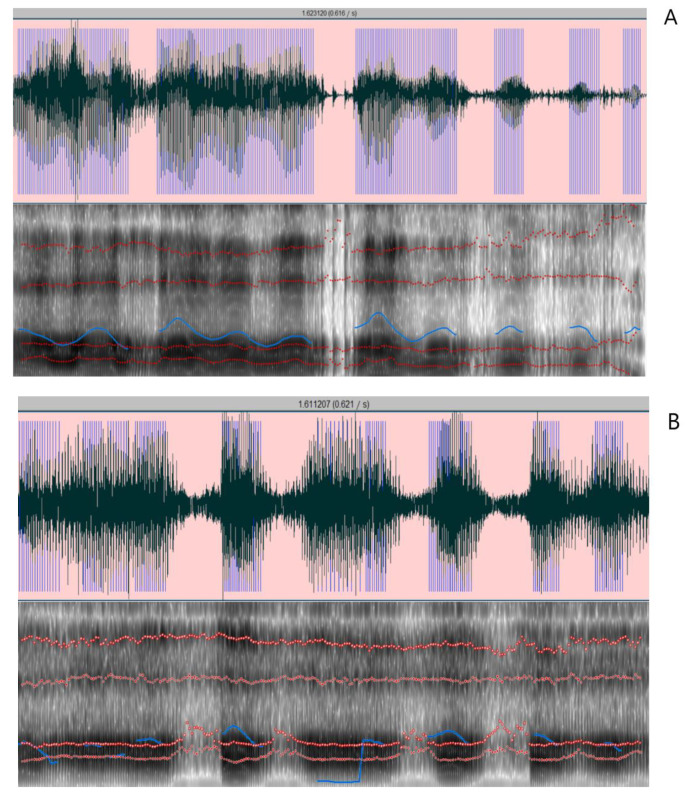
(**A**) The duration of the selection: 1.623120 s, Pitch: Median pitch: 199.863 Hz, Mean pitch: 200.121 Hz, Standard deviation: 17.163 Hz, Minimum pitch: 167.703 Hz, Maximum pitch: 247.906 Hz; Pulses: Number of pulses: 243, Number of periods: 237; Voicing: Fraction of locally unvoiced frames: 23.699% (41/173), Number of voice breaks (interrupted blue line): 5, Degree of voice breaks: 24.702% (0.426674 s/1.727257 s); Jitter: Jitter (local): 2.035%; Shimmer: Shimmer (local, dB): 0.714 dB; Mean harmonics-to-noise ratio: 8.846 dB. (**B**): The duration of the selection: 1.623120 s, Pitch: Median pitch: 179.098 Hz, Mean pitch: 169.509 Hz, Standard deviation: 35.773 Hz, Minimum pitch: 92.742 Hz, Maximum pitch: 218.821 Hz; Pulses: Number of pulses: 139, Number of periods: 129; Voicing: Fraction of locally unvoiced frames: 42.138% (67/159), Number of voice breaks (interrupted blue line): 8, Degree of voice breaks: 46.935% (0.756215 s/1.611207 s); Jitter: Jitter (local): 2.269%; Shimmer: Shimmer (local, dB): 0.915 dB; Mean harmonics-to-noise ratio: 4.313 dB.

**Figure 4 jcm-11-03453-f004:**
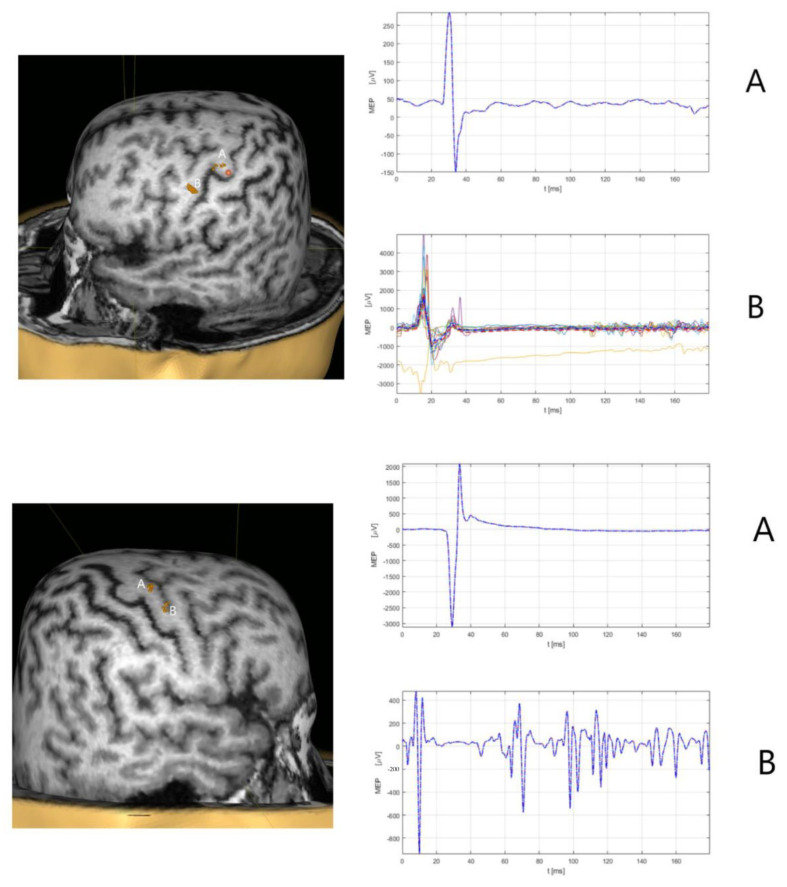
Transcranial magnetic stimulation (TMS) of the primary motor cortex for the upper extremity and laryngeal muscles. (**Upper**) TMS mapping of the left primary motor cortex for hand muscle representation (APB) and cricothyroid muscle representation with recording motor evoked potentials from the right-hand muscle (**A**), and corticobulbar motor evoked potentials from the right cricothyroid muscle (**B**). The latency of motor evoked potentials from the hand muscle is 23.61 ms with a peak-to-peak amplitude of 603.07 µV. The mean latency of corticobulbar motor evoked potentials is 11.67 ± 1.7 ms, and the cortical silent period (CST) duration of 89.54 ± 21.9 ms. (**Lower**) TMS mapping of the right primary motor cortex for hand muscle representation (APB) and cricothyroid muscle representation with recording motor evoked potentials from the left-hand muscle (**A**), and corticobulbar motor evoked potentials from the left cricothyroid muscle (**B**). The latency of motor evoked potentials from the hand muscle is 22.79 ms with a peak-to-peak amplitude of 106.07 µV. The mean latency of corticobulbar motor evoked potentials is 11.67 ± 1.7 ms, and the cortical silent period (CST) duration of 44.78 ± 6.9 ms.

**Table 1 jcm-11-03453-t001:** Literature overview on standard BoNT therapy and invasive brain modulation treatment effects.

Literature	No Subjects	Age	Sex (M/F)	Laryngeal Dystonia (Adductor/Abductor)	Clinical Presentation	Medical Treatment	Medical Treatment Outcome
Santos et al. [77]	1	61	F	Adductor	Tense voice, vocal tiredness, breathy voice, laryngeal pain, loss of voice extension, lack of frequency control	5U of type A Botulin Toxin (Botox) in the left thyroarytenoid muscle	Increased respiratory capacity and maximum speech time
Evidente et al. [78]	3	74 71 65	F F M	Adductor	Essential hand tremor with laryngeal dystonia	Bilateral ventralis intermedius (VIM) DBS	Could easily phonate with no vocal tremor, improvement of USDRS scores post-DBS compared to pre-DBS, and with stimulator on compared to stimulator off
Krüger et al. [79]	2	85 73	F	Adductor	Essential limb tremor with laryngeal dystonia	Bilateral ventrointermediate (VIM) nucleus DBS of the thalamus	Unanticipated improvement of their SD symptom; powerful unilateral benefit in both patients; hand dominant related probably
Poologaindran et al. [80]	1	79	F	Adductor	Right upper limb tremor with laryngeal dystonia	Left ventral intermediate (VIM) nucleus of the thalamus	Significantly improved SD vocal dysfunction compared with no stimulation (DBS off), as measured by the USDRS and VRQOL
Stewart et al. [81]	60	6078	42 F 18 M	Adductor	Roughness, strain-strangled voice quality, and increased expiratory effort	BoNT	Subjects reported benefit from BoNT injections, and had self-selected to return for continuing BoNT management of their voice symptoms when the benefits of the BoNT injections had diminished

Abbreviations: Unified Spasmodic Dysphonia Rating Scale (USDRS); voice-related quality of life (VRQOL).

## Data Availability

Data is available on request from the corresponding author.

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
