# Peer review of "The Patho-Neurophysiological Basis and Treatment of Focal Laryngeal Dystonia: A Narrative Review and Two Case Reports Applying TMS over the Laryngeal Motor Cortex"

_jcm, 2022, doi:10.3390/jcm11123453_

Round 1

Reviewer 1 Report

This manuscript summarizes the neurophysiological and pathological basis of focal LD and some current and promising treatments. Meanwhile, the manuscript discusses two case reports applying TMS over the laryngeal motor cortex. There are also the following suggestions that could be considered:

1.     For articles that combine review and case report, the arrangement of sections is important. For this manuscript, to exaggerate, the sections of review and the cases are enough to be divided into two articles. Some parts of narrative text, such as the method of injection of BoNT, are too detailed. Summarize some of the less important statements and focus more on the discussion to make the two parts more connected.

2.     The first patient suffered some severe side effects after taking medication, so the part of Pharmacological treatment possibilities and effectiveness in dystonia treatment can be further discussed. Indications, dosage and the side effects of treatment are all worth discussing.

3.     Page 9 line 339-340 “Application of low frequency (inhibitory) rTMS to the LMC might decrease the over-activation of the laryngeal muscles.” Are there any references to support this view? If so, is there any specific difference between this promising treatment and the TMS used by both patients for the excitability determination? Is it just the frequency difference? At the same time, it might be better to add a brief introduction to TMS as an evaluation tool. 

4.     Table 2: The text in the table can be summarized to make the table more concise. Meanwhile, the "Speech Therapy" column of the table has only one content, which is less related to the topic of the table, so would it be better to remove this column?

5.     Some discussion in this manuscript needs some relevant references to be more convincing, such as line 107-109, line 132-158, line 245-246, line 272-277.

6.     Page 14 line 515: a short introduction of autologous fat injection may be better.

7.     Some minor errors in the text need to be corrected, such as an extra space in line 103, an extra “-” in Table 2 (-could easily phonate). 

Author Response

We thank the honorable reviewer for the time spent reviewing our paper. Also, we are grateful for positive comments and suggestions, which helped us to improve the resubmitted paper significantly. In the resubmitted paper, we address every reviewer's comment and author’s response to each of the reviewer's concerns, and the description of the activities performed for addressing every concern can be found below. Responses to raised comments are attached in the word document.

Reviewer 2 Report

Review of manuscript titled “patho- neuro-physiological basis and treatment of focal laryngeal dystonia: a narrative review and two case reports applying TMS over the laryngeal motor cortex”. The authors present patho-neuro-physiological basis of laryngeal dystonia. Their paper is interesting, but needs improvement in some aspects.

6. Case report of LD patients

6.1

Which part of the primary motor cortex was stimulated?

Perhaps sectioning would make the reading easier.

Did the TMS improve the patient’s symptoms?

Why was medication treatment introduced in 2021? Lack of TMS effect?

A visual representation of the rTMS area would be interesting.

For both cases it’s difficult to follow if the rTMS improved the patients symptoms?

If so, how long did the effect last?

Conclusion

This section is extensive and contains repetitions.

Author Response

(The authors gave the same response as above.)

Round 2

Reviewer 1 Report

The authors have carefully addressed the comments and made the manuscript much more fluent in logic flow than the last version.